# DATA CURATION FOR LARGE-SCALE DETECTION PRE-TRAINING

## ABSTRACT

Large multimodal datasets gathered from the internet have been a key driver of progress in recent image-text models such as DALL-E, CLIP, and Flamingo. However, structured prediction tasks have not seen the same benefits, as noisy fine-grained annotations do not exist at web-scale. In this paper, we aim to extend the gains enabled by web-sourced training sets to the problem of object detection. First, we show that data curation for grounding and localization necessitates its own approach: filtering methods which produce good datasets for image classification with CLIP models (e.g., the image-text similarity filtering from LAION-5B) do not yield better object detectors. Instead, we introduce new detection-focused filtering methods that match or outperform existing object detectors pre-trained on supervised detection pretraining datasets. When trained on 102.4M images from the 12.8B image DataComp pool in a weakly supervised manner, our new filtering method matches the performance of a detector pre-trained on Object365, the largest fully-annotated detection dataset. In addition, our filtering approach shows good scaling with training set size and can be combined with Object365 to yield further improvements. To aid further research in this area, we release a 2.8B image subset of DataComp-12.8B pseudo-labeled with region proposals and detection bounding boxes.

## 1 INTRODUCTION

Recent breakthroughs in machine learning have been driven by improvements in scaling methods. A crucial aspect of this scaling is the ability to leverage large quantities of web-scale data to train models without any explicit human supervision. This approach has been successful for large language models OpenAI (2023); Touvron et al. (2023a;b); Elsen et al. (2023) and vision-language models Alayrac et al. (2022); Radford et al. (2019); Aghajanyan et al. (2023). Key to this is the existence of *noisy human supervision*, for example image alt-text for training vision-language models or the natural structure of text for training large language models. While these approaches work well for many high level tasks, structured prediction vision tasks such as object detection, segmentation, and tracking still rely strongly on supervised human annotations for performance (Zong et al. (2022); Zhang et al. (2023); Gupta et al. (2019)) and do not enjoy the same benefits of data-scaling as large language models or vision language models. This is because noisy versions of such structured prediction annotations do not exist at web-scale.

We bridge this divide by studying how data distributions can enable better data scaling for structured prediction vision tasks. In this context, we study the following question: *How can we curate scalable datasets for structured prediction?* We look at this problem in the specific context of object detection, an important problem in computer vision with a rich history of methods and scaling. Current detection methods that utilize web-scale data for object detection generally use the same image-text datasets as those used for representation learning methods such as CLIP or ALIGN Radford et al. (2019); Minderer et al. (2023).

We begin by investigating whether existing web-scale datasets are useful for detection. To do this, we fix a weakly supervised pre-training procedure which incorporates filtered unlabeled image datasets. We pseudo-label a given dataset with a detector trained on LVIS (Gupta et al., 2019), a human-labeled long-tailed detection dataset. We use these pseudo-labels to pre-train a detector which we then fine-tune on LVIS. This allows us to compare datasets by using performance on

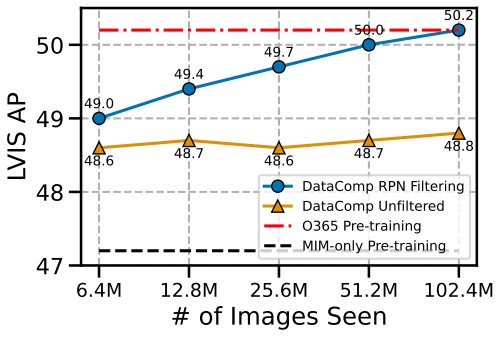 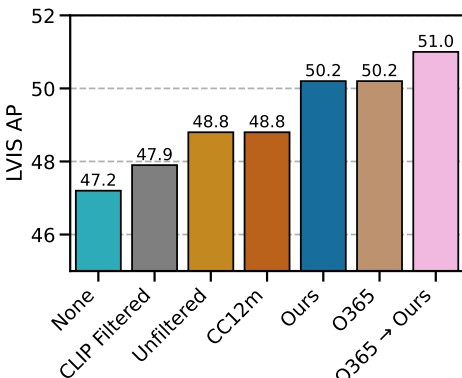

Figure 1: **Detection-aware filtering yields scalable datasets.** We evaluate the quality of a dataset for detection using the procedure detailed in Section 3.3. The black line on **(left)** and "None" on **(right)** shows the performance of the pseudo-labeler. **(left)** shows that without detection-aware filtering, performance does not scale with parameter count. **(right)** Shows that we can match Objects365 pre-training with a *fully un-annotated dataset*, and that our method stacks with Objects365 pre-training.

LVIS as a proxy. When we evaluate datasets filtered for CLIP performance (Gadre et al., 2023), we find that they perform *worse* than unfiltered datasets in this setting.

Based on negative results for existing filtering methods, we introduce a new detection-focused curation method. Our method is based around filtering for images which show more complex scenes using the pseudo-labeling model itself. We show that our simple method outperforms the unfiltered and CLIP filtered pools by large margins (+1.4 AP and +2.3 AP on LVIS respectively), and that it can be easily scaled to larger datasets and detection models. Furthermore, when controlling for dataset size, we show that we can construct a filtered dataset that outperforms CC12M, a dataset commonly used for detection pseudo-labeling.

Moreover, we demonstrate that our method benefits from data scaling, whereas using unfiltered data does not. With our filtering, we show that we can match Objects365 pre-training, a the largest entirely human annotated detection dataset (600k images), using completely machine generated labels and also that our method can be complementary to this dataset (yielding a 0.8 AP improvement). Finally, we demonstrate that our performance improvements scale with model size to large detection models (ViTDet-L, (Li et al., 2022)).

We hope that our work will be used as a starting point for data curation methods for fine-grained tasks. We release a 2.8B subset of DataComp 12.8B annotated with pseudo-labels and region proposals to aid with further work in this area.

## 2 BACKGROUND & RELATED WORK

### 2.1 OBJECT DETECTION

Object detection has remained a fundamental computer vision problem over the years, and is a basic building block of many other vision problems such as object tracking, instance segmentation, panoptic segmentation and keypoint detection Ren et al. (2015); He et al. (2017); Wu et al. (2019); Redmon et al. (2015); Kirillov et al. (2018); Carion et al. (2020). There are many approaches to object detection, ranging from multi-stage fully convolutional closed-vocabulary detectors, single stage transformer based open vocabulary detectors. Often these methods all use the strongest vision representation available as a backbone, which was ImageNet-based CNNs for older models such as Faster-RCNN and now is usually a CLIP-trained visual transformer (ViT).

Detection tasks evaluated by their mean average precision (mAP) on COCO and LVIS Gupta et al. (2019); Lin et al. (2014). Both datasets use the same pool of images, but vary in the classes annotated. COCO focuses on 80 common objects, whereas LVIS focus on long tail detection and covers

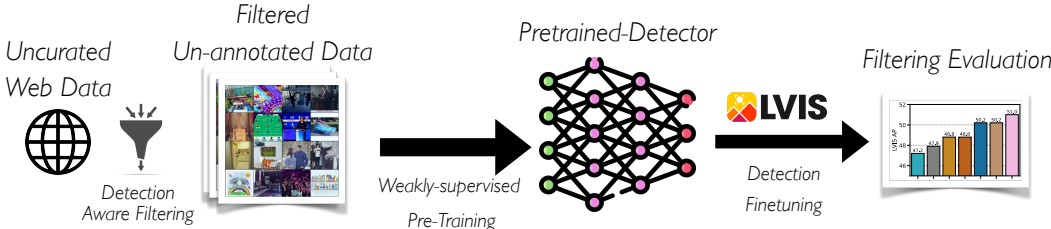

Figure 2: Dataset construction and evaluation overview. We investigate several filtering methods of large un-annotated & uncurated datasets, and evaluate them by training and testing a detector. The resulting boost of training on the filtered data vs on the unfiltered data is 1.4 mAP.

over 1200 objects. LVIS, the primary benchmark we use in this work, was constructed to include the long-tail of visual categories present in everyday life. As a result, it has several splits for evaluation depending on the rarity of the category in the training dataset. This makes it a good target for large-scale data augmentation. A common practice for state of the art detectors is to precede the LVIS/COCO finetuning phase with a detection pretraining phase on a large object detection dataset such as Objects365 (Shao et al. (2019)), this pretraining procedure usually ilicits 2 to 5 points of improvement in final mAP.

## 2.2 WEAKLY SUPERVISED AND OPEN-VOCABULARY OBJECT DETECTION

A recent focus on *open-vocabulary* object detection (Minderer et al. (2022); Zhang et al. (2022)) places an emphasis on evaluating on classes with unseen detection labels (though those concepts may have been seen during Image-Text pretraining). Many of these methods propose mechanisms for training on weakly-annotated (e.g. image-level annotation) or unlabeled data. Zhou et al. (2022) proposes utilizing image-level labels from ImageNet-21k to augment the training pipeline. ViLD Gu et al. (2021) proposes distilling from CLIP features to get an open-vocabulary detection model in the absence of open-vocabulary detection labels. Minderer et al. (2023) focus on scaling open-vocabulary detection to web-scale by pseudo-labeling a webscale dataset with a smaller open-vocabulary model and implementing fast training procedures to improve speed of training. Our work is complementary to these approaches, as many of them rely on weak supervision across large unlabeled datasets and our filtering methods are directly applicable in these settings. None of these works directly study the data distributions of the unlabeled data they use to augment their training.

For our work we will build on top of EVA (Fang et al. (2022; 2023)) which is a closed-vocabulary high performance detection pipeline that builds on top of a custom detection optimized CLIP backbone, followed by a multi-stage Cascade Mask R-CNN detector training procedure. We opt for EVA for its high performance on LVIS and other downstream tasks and reproducible pipeline.

## 2.3 DATASET CONSTRUCTION

High quality datasets have been the lifeblood of all machine learning tasks for many decades Highleyman & Kamentsky (1959). Yet until recently the core *purpose* of a dataset was tied to a specific end task, hand written digits were designed specifically for digit recognition and not expected to help with other recognition such as dog classification. However with the advent of representation learning on large diverse datasets starting with ImageNet and now massive image-text datasets such as LAION-5B, DataComp or WebLI (Schuhmann et al. (2022); Gadre et al. (2023); Chen et al. (2023)), we expect a single large dataset to improve performance on many diverse tasks.

A closer look at the dataset construction methodology of the large image-text datasets and detection specific datasets reveals an inherent tension. Datasets designed for representation learning focus on including samples with high *alignment* between the image and alt-text often using a pretrained Contrastive Image Language pretrained (CLIP) model. Whereas datasets designed for detection such as MSCOCO tend to optimize images that contain many objects in a non-iconic setting (not large, centered objects) Chen et al. (2015). Since alt-text is often a short description of an image, an alignment based dataset will fail to include the precise images that lead to good object detectors.

## 3 DATASET CREATION AND EVALUATION METHODOLOGY

We propose a scalable method for finding large and high-performing datasets for detection. The design of this method is based on three choices: **(1)** the initial pool of data; **(2)** the filtering method to efficiently identify a suitable subset of this data for weak detection supervision; **(3)** evaluation of this subset of data for weakly supervised detection quality (see Figure 2).

### 3.1 INITIAL DATASET POOL

We use DataComp-12.8B (Gadre et al., 2023) for our initial large-scale pool. DataComp-12.8B is a web-scale image-text dataset used for studying how data distributions affect the performance of image-text models, specifically CLIP. This pool is an *unfiltered* sample of web data, giving us a lens into how to perform detection-aware data curation at web-scale. We chose this dataset for its size, diversity, public availability, and depth of documentation. In particular, its size and diversity are key factors, as they allow us to test the scaling properties of a variety of more aggressive filtering methods. Further, given its use with CLIP training, we can examine the relationship between datasets optimized for CLIP performance and those optimized for detection performance.

To improve the feasibility of this study, we focus on the subset of DataComp-12.8B with English captions, images of at least a certain size defined as $\min(\text{width}, \text{height}) \geq 200$, and aspect ratio $\frac{\text{width}}{\text{height}} \geq 0.3$, leaving us with a subset of 2.8B images with some minimum degree of image quality. This forms our initial pool on which we evaluate different filtering methods.

### 3.2 DATA FILTERING METHODS

In order to design a data filter, we need to take several aspects in consideration. Specifically, we need to weigh both the *selectiveness* and the *computational cost* of the approach. If a filtering method is highly selective, then we need to process large number images to produce a dataset of sufficient size. Similarly, if a filtering method is expensive to run, either because it requires human raters or running an expensive filtering network, then this also can limit the size of the final dataset. We would like to evaluate filtered pool sizes of 12.8m and 128m, corresponding to small and medium pool sizes in Gadre et al. (2023). This allows us to test the scaling properties of our method. However, this limits the selectiveness of our filtration methods to at least 5% of the initial pool (128m/2.8b). We do not have such hard constraints on the computational cost of the methods, however in general we study cheaper and more scalable filtering networks as discussed in the next section. Further, since we have to scale to over 100M images, a human-in-the-loop data annotation pipeline like in Segment Anything (Kirillov et al., 2023) is infeasible.

**Preliminaries for detection-aware filtering.** We propose using an inexpensive filtering detector trained on a large number of generic classes. We assume that this detector can produce object proposals and their associated labels. An example of these are two-stage detectors, such as Faster-RCNN, that produce object proposals in a first stage and classify these proposals into object classes in a second stage (Girshick, 2015; Li et al., 2022). For such a model, denote its output for an image by $(\{p_b, o_b, l_b, c_b | b \in B\})$ where for each object proposal $b$ among all proposals $B$ the quantities $p_b$, $o_b$, $l_b$ stand for the bounding box, the objectness confidence, the most confident object label and its label confidence. The first two quantities are produced by the first stage of the detector, often referred to by Region Proposal Network, while the second two quantities are produced by an object classifier.

**Region Proposal Network (RPN) Filtering.** A natural criterion for filtering is to identify images that might contain at least several objects. We can do this by utilizing the above filtering detector and retain images for which the detector produces several object proposals with sufficient confidence. We provide pseudocode for this filtering approach in Appendix C. We ablate the minimum box count and minimum objectness threshold in Fig. 8. Note that we use the objectness confidence and not the object classification confidence, as the former has been shown to be more robust to distribution shift than the classification component (Zhou et al., 2022; Gu et al., 2021), making it an excellent model for data filtering.

**Object Class Entropy Filtering.** Another object-based criterion is to retain images that have a diverse set of object classes. This can be defined using entropy over the frequency of detected

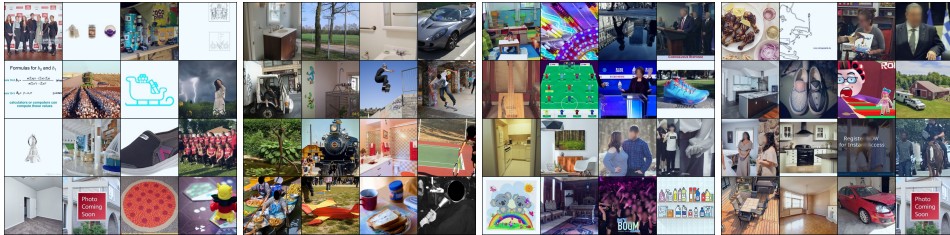

Figure 3: Qualitative examples of (left to right) **(a)** DataComp unfiltered, **(b)** MSCOCO, **(c)** DataComp with our region-proposal network detection-based filtering, **(d)** Datacomp with our label entropy detection-based filtering. Our detection-aware filtering methods yield significantly more complex and cluttered images, closer to what detection benchmarks evaluate against.

objects per image, computed using our pseudo-labeler. We threshold at an entropy value denoting the 75th percentile of our overall image entropy distribution. We provide pseudocode for our filtering logic in Appendix C.

The above method does not use the objectness score, and is therefore more affected by distribution shift. However, since we are only using it as a rough filtering metric with a low confidence threshold, it provides an interesting data distribution to study.

**CLIP Filtering** As a baseline, we also consider CLIP-based filtering as proposed in Schuhmann et al. (2022; 2021). It attempts to identify images that are well-aligned with their accompanying text, stipulating that such data points are of higher quality. In particular, CLIP filtering (Schuhmann et al., 2021) uses the CLIP model from Radford et al. (2021) to score image-text pairs from Common Crawl using the cosine similarity between extracted image and text embeddings. LAION-400M, for example, uses images over with CLIP score over 0.28 to construct their final dataset. Pseudocode for CLIP filtering is provided in Appendix C.

### 3.3 DETECTION PSEUDO-LABELING AND DATASET EVALUATION

In order to compare filtering methods, we need a consistent way to utilize arbitrary unlabeled image data for detection. In short, we propose using this unlabeled data for pseudo-labeling, then using fine-tuning performance to evaluate the psuedo-label quality. The closest analogue for this weakly-supervised training procedure in literature is detection self-training (Zoph et al., 2020).

**Pseudo-Labeling.** We start with a detection model fine-tuned on a particular task. For every image in our image-text corpus, we use this model to produce bounding box pseudo-labels. Each pseudo-label contains object category information, confidence information (i.e. how likely according to the model that it is an actual category), and localization information in the form of a bounding box. We then threshold bounding boxes on confidence (see Appendix B) and include images which have at least one detection.

**Detector training and evaluation of detection datasets** We use the resulting pseudo-labeled dataset to train a detector and evaluate its performance. Since we are primarily concerned with understanding the best strategy to construct a detection dataset, we keep the detector training and evaluation methods fixed and only vary the pseudo-labeled dataset. Thus, the evaluation of the final detectors can be used to analyze the effect of the different filtering procedures we propose.

The training procedure is a two-step approach, where we initially pre-train a detector on the filtered and pseudo-labeled data, and subsequently fine-tune it on a downstream task. The downstream data is fixed across our experiments, while the pre-training pseudo-labeled data is what we curate.

**Weight-ensembled evaluation** At test time, we include a step to keep some of the robustness properties of the pre-trained model. We average the weights of the pre-trained and fine-tuned model with a mixing coefficient $\alpha$. For each corresponding weight in the pre-trained and fine-tuned network $w_i^{pt}$ and $w_i^{ft}$ respectively, we compute a new weight $w_i^o = w_i^{pt}(1 - \alpha) + w_i^{ft}(\alpha)$. For weights in the fine-tuned network that do not exist in the pre-trained network, we keep the fine-tuned version. This has been shown to improve robustness when fine-tuning foundation models (Wortsman et al., 2022). We see interesting empirical improvements when applied here, and include an ablation of this parameter in the Appendix.

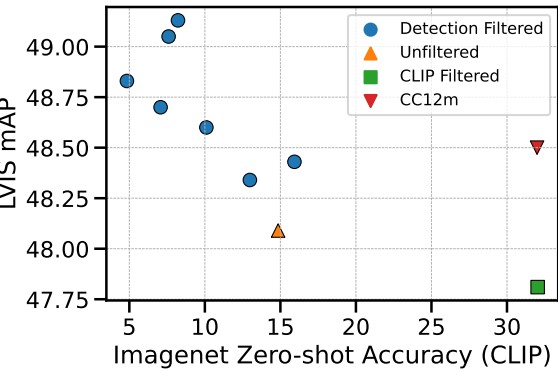

Figure 4: Relationship between CLIP performance on a suite of image-text benchmarks (Gadre et al., 2023) versus LVIS (Gupta et al., 2019) AP on various dataset distributions of size 12.8M. We see that in general, CLIP performance is a poor predictor of detector performance, and in fact negatively correlates with detector performance. Therefore it should not be used to filter datasets for detection.

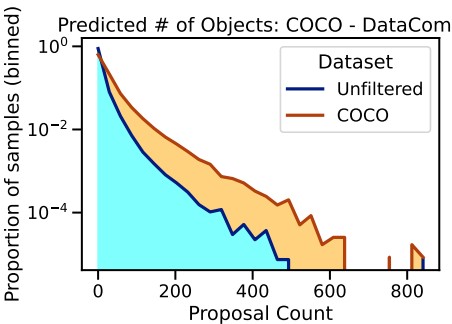
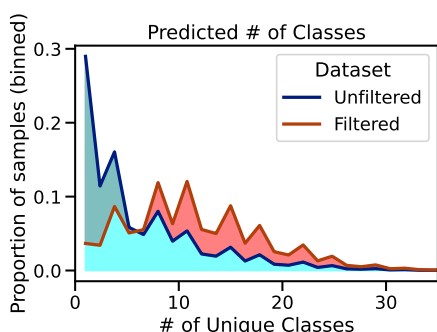

Figure 5: **(left)** Difference between predicted object count. Generally detection benchmarks have more diverse scenes. **(right)** After filtering, DataComp has significantly more unique classes per image signifying a more diverse images in our data distribution.

Using a fixed detector training and testing procedure to evaluate a training dataset has a connection with the dataset filtering benchmarks discussed in (Gadre et al., 2023; Schuhmann et al., 2022), where the datasets are evaluated with respect to zero-shot performance on downstream tasks. Note that since those works are comparing *zero-shot* performance, and we are comparing *fine-tuned* performance, the gap in performance between the worst and best methods will be necessarily smaller. Further, perform hyperparameter optimization on LVIS fine-tuning to ensure that the only variable is pre-training dataset quality, and not quality of fine-tuning. We discuss this further in Appendix D.

## 4 EXPERIMENTS

Our key experimental results can be summarized as follows: **(1)** Without filtering, pseudo-labeling on web-collected data exhibits minimal improvements with scale, **(2)** With the simple filtering methods discussed previously, we get substantial performance improvements with scale, matching or beating Objects365 pre-training performance, and **(3)** improvements with self-training stack with other forms of detection pre-training.

**Training details** For our detection architecture, we use a Cascade RCNN (Girshick, 2015) with a ViTDet backbone as proposed in Li et al. (2022) and modified in Fang et al. (2023). All of our models begin with an MIM pre-trained backbone from Fang et al. (2022). We do all of our training at 1024 by 1024 resolution with a ViTDet-B backbone unless otherwise stated. The model we use to pseudo-label our datasets, unless otherwise stated, gets 47.3 AP on LVIS, corresponding to no pre-training dataset in Table 1. We sample with replacement during pre-training, so whenever we state "number of examples seen" this does not equate to the *unique* number of examples seen. Full architecture and training details can be found in Appendix A.

### 4.1 CLIP PERFORMANCE DOES NOT PREDICT DETECTION PERFORMANCE

Here we study the relationship between CLIP performance and detection performance. Our setup allows us to evaluate detection performance for arbitrary datasets via weak supervision. Moreover,

Table 1: Comparison of different filtering methods for LVIS AP in the *12.8m pool size* regime. "r", "c", and "f" correspond to rare, common, and frequent classes respectively, and denote different splits of LVIS proposed in Gupta et al. (2019). We see that our detection-aware filtering methods produce datasets that outperform all others at this data scale.

| Pre-training Dataset | AP | APr | APc | APf |
|---|---|---|---|---|
| None | 47.3 | 37.2 | 48.1 | 50.8 |
| CLIP Filtering | 47.8 | 36.6 | 49.5 | 50.9 |
| DataComp Unfiltered | 48.7 | 37.4 | 50.3 | 51.8 |
| CC12m | 48.8 | 37.4 | 50.4 | 51.9 |
| Entropy Filtering | 49.2 | 39.7 | 50.3 | 52.1 |
| RPN Filtering | 49.3 | 39.4 | 50.4 | 52.4 |

since our detection filtered datasets are derived from DataComp, every training example also has associated noisy text labels, allowing us to train CLIP models on them. We take all the datasets used across all of our experiments and those in our ablation studies (Figure 8) of size 12.8M and train CLIP models for 128M images seen (further details in Appendix A). We then evaluate our method on benchmark evaluation datasets proposed in Gadre et al. (2023) and compute average performance. Our results are summarized in Figure 4. We see a clear negative correlation between detection filtered methods and CLIP performance, lending credence to our hypothesis that more complex images (e.g. images with many objects) are less likely to be well described, and therefore are worse datasets for CLIP.

### 4.2 UNFILTERED DATASETS FOR PSEUDO-LABELING

In Figure 1, we show key result (**1**): without filtering, pseudo-labeled training does not scale. In this figure, we plot the number of images seen during pre-training by the LVIS AP after fine-tuning. Ideally, when increasing the pre-training cost, we would like to see better fine-tune detector performance. Surprisingly, for unfiltered data, this is not the case.

Digging into the dataset itself, we see why this happens. In Figure 3 **(a)**, we show a random, uncurated set of examples from the DataComp dataset. We see that many of these images are product images or screenshots of other webpages. Essentially they do not contain many of the diverse scenes that many detection benchmarks require, as shown in the same figure **(b)**. To demonstrate this more quantitatively, we use our pseudo-labeler on the images of COCO and unfiltered DataComp respectively to get region proposals for each dataset. In Figure 8, we take the region proposals with objectness logit $\geq 5$ and plot binned proportions. We see that the unfiltered dataset is composed of far fewer images with a large number of proposals. Qualitative examples of this can be seen in Figure 3 **(c)**. We hypothesize that this lack of diverse images is what leads to poor quality pseudo-labeling, contributing to poor performance. We analyze this hypothesis further in Section 4.3.

### 4.3 OPTIMIZING FILTERING FOR DETECTION

Here we examine the three filtering methods described in Section 3.2. We have two main conclusions: **1.** Using CLIP filtering (an image-text filtering method) is *worse* for detection than not filtering at all, **2.** Detection-aware filtering methods yield significant improvements. For this experiment, we construct 12.8m sized datasets by sampling from DataComp-12.8B using each of the corresponding filtering methods. After pseudo-labeling, we pre-train our detector on these pools sampling $6.4 \times 10^6$ images with replacement (exact details in Appendix A).

Our results are summarized in Table 1. The results are clear both in overall AP and across the various rarity splits provided by LVIS. Our detection-based filtering methods provide an improvement over both unfiltered and CLIP filtered datasets of $0.6$ AP and a full 2 AP over the baseline. Interestingly, CLIP filtering actually *harms* downstream performance. It has been shown that CLIP has poor spatial understanding of scenes (Subramanian et al., 2022) and is good at picking up on text in images (e.g. optical character recognition) (Radford et al., 2021), to the point where text-based attacks are feasible. This leads to poor image selection for detection models, as we get few complex scenes and many screenshots.

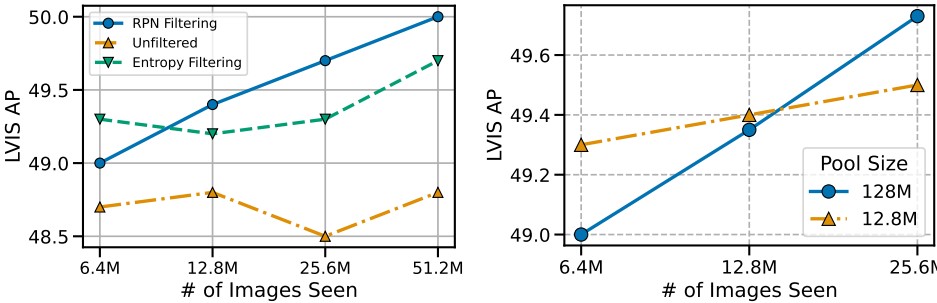

Figure 6: **Scaling trends of different filtering methods. (left)** With a dataset pool size of 128m, entropy filtering exhibits good performance at low scale but inconsistent scaling. By contrast, RPN filtering exhibits consistent log-linear scaling. **(right)** We show that pool size is crucial to data performance scaling. When the number of images seen during training exceeds the pool size, performance improvements slow.

### 4.3.1 CC12M

Compared to DataComp 12.8b, which is essentially an unfiltered pool of image-text pairs from Common Crawl, CC12m is significantly more curated. One key component of this is their image-text filtering, where they use the Google Cloud API to apply text tags to the image, and filter based on how many text components match between the caption and these predicted tags. This restricts us to only using data for which there are good corresponding captions. We hypothesize that more complex scenes are less likely to be reliably described by alt-text (a topic covered further in Section 4.3). We see in Figure 1 **(right)** that CC12m performs approximately the same as unfiltered DataComp 12.8m when used for pseudo-labeling, achieving 48.7 AP on LVIS.

### 4.4 SCALING

Here we study the scaling properties of our detection-based filtering methods, using the unfiltered pool as a baseline. The key takeaways are that: **(1)** Different filtering methods show different data scaling trends and **(2)** RPN filtering exhibits approximately log-linear scaling with pre-training datasets size. To demonstrate proper scaling, we extract filtered pools of size 128m from DataComp 12.8b, rather than 12.8m as in the previous section. We show that this matters for studying scaling trends in Figure 6.

**Scaling properties of different filtering methods** In Figure 6 **(left)**, we show that not all filtering methods exhibit the same scaling properties. Label entropy-based filtering performs better at 6.4M images seen (49.3 vs 49.0 AP) but worse at higher data regimes (49.7 vs 50.0 AP at 51.2M examples seen). We also see an interesting "delayed scaling" trend where the model only begins improving with scale after 12.8M images seen. By contrast, RPN filtering exhibits a more consistent approximately log-linear relationship with scale, improving consistently with more pre-training data.

**Pool size matters for scaling** In Figure 6 **(right)**, we show RPN filtering with two different pool sizes, 12.8M and 128M. Recall that during training, we draw samples with replacement from the data pool, so we can see the same example more than once in a single "epoch". We see that with pool size 12.8M, we get negligible performance improvements with increased training iterations, presumably because we have saturated the training pool. However, with 128m images, for the same training budget, we see consistent performance improvements.

**Comparison to Objects365 pre-training.** Objects365 (Shao et al., 2019) is currently one of the strongest forms of supervised detection pre-training, with over 600k human annotated images. In Figure 1 **(right)**, we show that introducing Objects365 pre-training to the EVA-02 pipeline at ViTDet-B scale improves performance by 3.0 AP over vanilla pre-training. In Figure 1, we show that we can match this performance with *no extra annotated data* using our pseudo-labeled RPN filtered pool. Further, by introducing our pseudo-labels after Objects365 pre-training, we can further improve performance to 51.0 AP, +0.8 AP above Objects365 pre-training.

| Pre-training Dataset | Re-tagged? | AP | APr | APc | APf |
|---|---|---|---|---|---|
| None | - | 60.5 | 53.2 | 62.2 | 61.8 |
| RPN Filtered | No | 60.3 | 52.7 | 61.8 | 62.2 |
| Entropy Filtered | No | 61.3 | 53.9 | 63.2 | 62.7 |
| RPN Filtered | Yes | 61.2 | **56.7** | 62.5 | 61.8 |
| Entropy Filtered | Yes | **61.4** | 55.0 | **62.9** | **62.4** |

Table 2: Our detection aware filtering methods can be used to improve performance at larger model scales. When we re-label our dataset with a higher performing model ("None" in this table), we can use these datasets in a weakly supervised fashion to improve performance. Interestingly, with entropy-based filtering, we do not need to re-tag to see performance improvements, meaning we are using labels from a model which gets 47.3 AP to improve a model which gets 60.5 AP.

## 4.5 SCALING TO LARGER MODELS

Here we study how well our methods transfer to larger models (ViTDet-L) at higher resolutions ($1536 \times 1536$). Our results in this setting are summarized in Table 2. For all experiments, we train for 12.8M images seen during detection pre-training. The re-tagged column shows whether or not we pseudo-label using the Cascade-RCNN with a ViTDet-L backbone (which gets 60.5 AP) or keep the original pseudo-labels from the ViTDet-B backbone (which gets 47.3 AP). This would be analogous to the setting at ViTDet-B scale. In general, we expect performance without re-tagging to be poor, since the model we use to pseudo-label gets only 47.3 AP, whereas with only backbone pre-training, our model gets 60.5 AP. This is born out in our RPN filtering without re-tagging results. Interestingly, it seems that we can pre-train with our entropy filtered dataset and still get some improvement without re-tagging. This demonstrates that our pseudo-labels themselves can be useful for detectors outside of filtration. Further, refining the bounding boxes by re-tagging provides uniform performance improvements.

## 5 CONCLUSION & DISCUSSION

In this paper, we present a procedure for determining the quality of a dataset for detection in the absence of ground-truth labels. To do this, we fix a weakly-supervised training algorithm based on pseudo-labeling which utilizes our curated dataset with respect to a downstream task, LVIS, and use performance on this task to determine the quality of this dataset. Through this evaluation procedure, we show that current methods of filtering data at scale (e.g. CLIP filtering) are often poor for detection. We build on this result to propose two detection-aware filtering methods, which greatly improve detection performance over uncurated data which are readily available on the internet (Schuhmann et al., 2022; Gadre et al., 2023). Finally, we show that we can construct datasets which match Object365 pre-training, the best *fully supervised* dataset for detection pre-training, and that our method is complementary to Objects365 pre-training.

We hope that our benchmark encourages future research in this area. In our work, we see that filtering is necessary to get good performance with data scale when relying on weak supervision. Therefore its likely that producing a "foundation model" level detector will rely on such data filtering procedures. This kind benchmark also readily extends itself to other fine-grained tasks which do not have large-scale labels, which is an important avenue of future work.

## 6 REPRODUCIBILITY

We plan to provide a full code release along with a release of the pseudo-labels we generated for a 2.8B image subset of DataComp-12.8B. Moreover, all the unfiltered images we use are part of public datasets (DataComp-12.8B and CC12M).

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

## A  TRAINING AND ARCHITECTURE DETAILS

In all cases we start with an MIM pre-trained backbone as described in Fang et al. (2023). We also start with a Cascade R-CNN, a two-stage detector proposed by Cai & Vasconcelos (2018).

### A.1  PSEUDO-LABELED DETECTION PRE-TRAINING

During pre-training, we use only the detector losses from Cascade-RCNN. We train with batch size 128, sampling with replacement from our pre-training dataset. Pseudo-labeled bounding boxes which have passed the confidence threshold are treated as ground-truth. Number of images seen in our plots is computed as batch size $\times$ iterations trained.

We train using the AdamW optimizer with the following parameters:

- Learning rate $6 \cdot 10^{-5}$
- beta 1 = 0.9
- beta 2 = 0.98

We use per-layer learning rate decay described by Li et al. (2022) with a decay rate of 0.8. Our learning rate schedule is a multi-step procedure gamma=0.1 and warmup. We warmup from 1/100 the max lr for 1/20th the total number of iterations and drop learning rate at 8/10ths and 9/10ths the total number of iterations respectively. Empirically this outperforms cosine decay with the same warmup and max lr.

**Objects365 pre-training**  We use the same details as above, training for 100k iterations. We noticed that training for more iterations has a negligible affect on performance.

### A.2  LVIS FINE-TUNING

We take great care to ensure that LVIS fine-tuning is as optimal as possible, so that the only effect on final performance is the pre-training dataset. To do this, we do a grid search along the following parameters, and use the best performing model for each baseline:

- Learning rate: 0.0001, 0.00006, 0.00001
- Weight-ensemble $\alpha$: 0.7, 0.75, 0.8, 0.85, 0.9
- Training duration: 10k, 25k, 50k iterations

### A.3  VITDET-L

All the details above apply to ViTDet-B. For ViTDet-L, the primary differences are that we use exponential moving average for pre-training and fine-tuning with decay rate 0.9999. For fine-tuning we also train longer, for 70k iterations. Since this scale is so expensive, we only compare learning rates 0.0001 and 0.00006.

### A.4  FILTERING METHOD DETAILS

**RPN Filtering.**  We use objectness threshold of 5 and a minimum box threshold of 10 for most of our experiments unless otherwise stated. These choices are explored further in Appendix E.

**Entropy Filtering.**  We threshold at entropy 2.0, corresponding to the 75th percentile of the image label entropy distribution, with a confidence threshold 0.4.

## B    CONFIDENCE THRESHOLD ABLATION

Figure 7 shows the result of a confidence threshold ablation on our unfiltered pseudo-labeled pool. We see that both too low and too high a confidence threshold are sub-optimal. Too low, and we include too many false positives in our dataset, generating noise. Too high, and we filter out far too many bounding boxes, changing the fundamental layout distribution of the annotations.

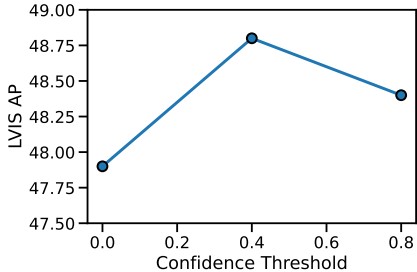

Figure 7: Confidence threshold ablation on DataComp Unfiltered. We see that both too low and too high a confidence threshold are sub-optimal.

## C    PSEUDOCODE

For each piece of pseudo-code, if it returns true on an image, then we reject the example.

### C.1    REGION PROPOSAL NETWORK FILTERING

```
def rpn_filter(image, proposals, min_confidence, min_proposals):
    count = len([p for p in proposals if p.confidence >= min_confidence])
    return not (count >= min_proposals)
```

### C.2    ENTROPY FILTERING

```
def entropy_filter(image, box_labels, thresh, all_classes):
    freq = {label: 0 for label in all_classes}
    for box_label in box_labels:
        freq[box_labels] += 1
    probabilities = [f / len(proposals) for f in freq.values()]
    return entropy(probabilities) <= thresh
```

### C.3    CLIP FILTERING

```
def clip_filter(image, text, threshold=0.28):
    # compute image and text representations
    image_features = clip.encode_image(image_input)
    text_features = clip.encode_text(text_input)
    # compute alignment
    dot_product = image_features.T @ text_features
    norm_a = image_features.norm()
    norm_b = text_features.norm()
    similarity = dot_product / (norm_a * norm_b)
    # filter by alignment
    return similarity > threshold
```

## D    EVALUATION OF PRE-TRAINED MODELS

Since our data curation evaluation procedure involves training on pseudolabels and then fine-tuning on LVIS for 50K iterations, the effects of the pre-training dataset can appear muted. This effect

| Pretraining Dataset | LVIS AP | Pascal AP |
|---|---|---|
| Unfiltered DataComp | 29.8 | 53.7 |
| RPN Filtered DataComp | 39.1 | 55.5 |

Table 3: Detector AP after training *only* on pseudo-labels without fine-tuning on an end task. We find that both on LVIS and Pascal evaluation AP are higher. This is another lens to show that these filtering methods produce higher quality datasets.

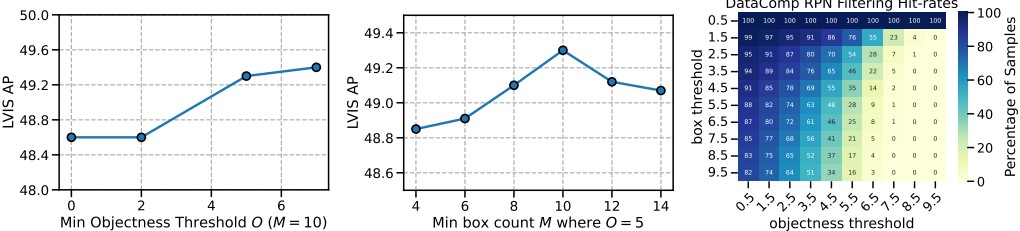

Figure 8: We show the importance of objectness threshold and minimum proposal count for our RPN filtering procedure. Given the hit-rates for our region-proposal filtering, we choose $O = 5$ and $M = 10$ to scale our method, which gives good performance and only filters 75% of our dataset.

has been shown in other models such as CLIP when observing quantaties such as ImageNet top-1 accuracy after finetuning the network on ImageNet (Cherti et al., 2022; Entezari et al., 2023). Thus *zero-shot* accuracy is often used to evaluate dataset quality for CLIP models (Gadre et al., 2023). For our problem we can evaluate the accuracy of the pretrained model *before* fine-tuning to measure the effect of our filtering intervention. We present our results for this in Figure 3.

## E    FILTERING HYPER-PARAMETER ABLATIONS

**Effects of Filtering Parameters** In Figure 8, we see how the different factors in RPN filtering, our best performing filtering method, affect downstream performance. The two main factors to study here are objectness threshold and the box count threshold, which we study at pool size 12.8m. We keep the bounding box confidence threshold fixed at 40% for consistency. In general, increasing the objectness threshold increases performance, whereas increasing the minimum object count (with a fixed threshold) saturates in performance around 8-10 boxes.

**Relationship between filtering aggressiveness and performance** These experiments are complicated by the fact that these two factors are entangled: changing the objectness threshold changes the number of predicted proposals per image. This is illustrated in Figure 8 (**left**), where we show a heatmap of the proportion of dataset exapmples accepted by a given combination of box count threshold and objectness threshold. We see that when we increase objectness threshold past 5, the number of proposal boxes drops off rapidly. This is why we do not use an objectness threshold of 7 in our experiments, even though it performs the best at this scale: its hitrate would be about 0.3%, making it difficult to scale datasets of this kind.

