# OpenReview forum: "Data Curation for Large Scale Detection Pretraining"
_ICLR.cc/2024/Conference — ICLR 2024 Conference Withdrawn Submission_

### Official Review · Reviewer_NGVY · 2023-10-26

**Soundness:** 2 fair
**Presentation:** 3 good
**Contribution:** 2 fair
**Rating:** 3
**Confidence:** 5

**Summary:**

The paper proposes two approaches to filter out unlabeled images that might not be helpful to the detectors. The proposed two approaches are simple and scalable: region proposal network filtering and object class entropy filtering. RPN-filtering keeps the images with many valid objects, and entropy-filtering keeps the images with a diverse set of objects. The paper evaluates the proposed methods on LVIS and finds that both RPN- and OCE-filtering outperform CLIP-filtering.

**Strengths:**

- The paper takes the first step to understanding the criterion of a good pre-trained dataset for detection pre-training.
- The proposed evaluation pipeline (weight-ensemble evaluation) is thoughtful, potentially leading to a more solid conclusion.
- The proposed filtered unlabeled dataset matches the Objects365 in terms of detection pre-training
- The proposed approach is very scalable and with little cost

**Weaknesses:**

**General**

- The paper limits the pre-training algorithm the study to weakly-supervised pre-training, which is not stated in the introduction.

**Experiments**

- The paper uses a popular dataset, LVIS, as the main dataset. While the proposed filtering approaches give slight improvements over CLIP-filtered dataset (+1.5 AP) and un-filtered dataset (+0.6 AP), it’s questionable if the proposed filtering approach apply to datasets like PASCAL which usually contains a lot less number of object per image. I suspect that the reason why RPN and entropy filtering works since it implicitly encode the domain knowledge in LVIS.
    - Two experiments/evaluations can make the paper stronger.
        1. Performing detection fine-tuning on other datasets can make the paper stronger. For example, first pre-train on a (LVIS-trained) RPN-filtered dataset and fine-tune on PASCAL, Cityscapes, etc.
            - Table 3 reports the number on PASCAL *without* detector fine-tuning. The results show that the improvements are limited as compared to LVIS.
        2. Split the LVIS into two sets and evaluate them separately: one with a small number of objects per image and another with a large number of objects per image.

**Questions:**

- Can the author further elaborate Table 3 in more detail. From my understanding to the paper, the pre-training stage is basically training a detector on a LVIS-labeled dataset. How do you evaluate this pre-trained detector on PASCAL without fine-tuning?
- The paper focuses on curating a dataset for detection pre-training. However, detector pre-training is not limited to weakly-supervised pre-training. There are other general vision pre-training methods. Can the author justify this point? For example, I am curious how the proposed filtering approaches work if self-supervised learning is used for pre-training.

---

### Official Review · Reviewer_hLdz · 2023-10-31

**Soundness:** 3 good
**Presentation:** 3 good
**Contribution:** 2 fair
**Rating:** 5
**Confidence:** 4

**Summary:**

In this paper, the authors present a new filtering algorithm used to curate a web-scaled dataset of images to improve object detection.
They start with a big pool of ~3B images and a detection model pretrained on LVIS. Then they produce pseudo-labels on the uncurated images and only keep images that reach a certain score depending on the number and the confidence of detected objects.
They compare their methods against the classical CLIP image-text filtering, the original uncurated dataset and an entropy based filtering on the number of predicted classes per image.
By pretraining on curated dataset created with their method, they reach better performances than the baseline when finetuning on LVIS.

**Strengths:**

- The paper is well written and the contribution is clear
- The experimental setup is well described
- The ablation studies are done with a grid search of hyperparameters for baselines and for the main method, which is necessary to fairly compare methods while finetuning.
- This is the first paper showing that the CLIP curation pipeline decreases object detection accuracy.
- The gain between the curated and uncurated pretraining is non trivial

**Weaknesses:**

- I think that showing that the CLIP filtering method is bad for object detection is a very good results, but the paper lack a real baseline. The CLIP filtering method is only trying to find good image and text pairs. In the meantime, the DINOv2 paper try to filter images based on their similarity with downstream task datasets (DINOv2: Learning Robust Visual Features without Supervision by Oquab et al. https://arxiv.org/abs/2304.07193). While it is difficult to compare against this paper as the authors don't provide the dataset, they use a pretty simple curation method based on the cosine similarity between self-supervised embeddings of a curated dataset (here LVIS) and a web-scaled noisy dataset (here DataComp) to filter their data, keeping a specific amount of k-NN per LVIS image to build a curated dataset. They also claim better semantic segmentation results than CLIP which might correlates with this paper's finding.
I think that using a simple image based retrieval filtering method on LVIS and then creating pseudo-labels is a reasonable baseline.

- While the reason why the method works is clear: the curated images have more objects so more weakly labeled signal during the pretraining, the ablation studies doesn't disentangle the fact that the detection model used to make pseudo labels is also trained on LVIS. By training the pseudo-labeler on LVIS, the images with the more objects are those that ressemble the most with images from LVIS. What happens if the pseudo_labeler is trained on another dataset like Object365?

**Questions:**

- I think that the paper has great potential, but the most interesting results are in Table 3. The quality of the curation process is hard to disentangle from the architecture performances while finetuning, but in Table 3 we can clearly see that the curated dataset outperforms the uncurated one.
    1) Could you include the performances of your bare pretrained models in ablation from table 1 and 2?

- It is not clear if the performances of the pseudo-labeler has a big impact on the curation process. In Table 2 it seems that a better pseudo-labeler is mainly improving rare class detection, while the curation method seems robust to the pseudo-labeler performances.
    2) Do you have any idea why it is helping only on rare classes?
    3) Why a model with less than 50AP is helping a finetuned model with more than 60AP? Is it because of the weight ensembling robustness increase or because of your filtering quality?

The following questions are optional and are not necessary for me to change my rating:

- Optional: Following with weakness #1:
   4) I know that doing a full baseline using a DINOv2 like approach is time consuming, but I would be satisfied with a smaller scale approach where you compute a small scale self-supervised filtering and compute the number of pseudo-labels classes as in Figure 5 (right). The number of classes should probably be in between the uncurrated one and your filtered method.

- Optional: Following with weakness #2: what happens if you use pseudo-labels with more classes? Does the method scale with the number of classes? The more classes, the higher the "RPN" score will be on some images.
    5) Do you have an ablation using a different pseudo-labeler trained on another dataset like Object365?
    6) Could you ablate the "scalability" of you method wrt. the number of pseudo-label classes by artificially removing some classes while filtering images? This could show that using better datasets to train the pseudo-labeler would help the filtering method to scale.

---

### Official Review · Reviewer_EkJ7 · 2023-10-31

**Soundness:** 2 fair
**Presentation:** 3 good
**Contribution:** 2 fair
**Rating:** 5
**Confidence:** 4

**Summary:**

This paper highlights the challenge of using web-sourced datasets for structured prediction tasks, specifically object detection, due to a lack of detailed annotations. The authors:

1. Identify that conventional data curation methods for image classification don't enhance object detectors.
2. Propose new detection-specific filtering methods that rival or surpass existing pretrained detectors.
3. Demonstrate that their method, when trained on 102.4M images from a 12.8B pool, matches the performance of a detector trained on the renowned Object365 dataset.
4. Will release a 2.8B image subset from DataComp-12.8B with pseudo-labels for community research.

In essence, the paper introduces new filtering techniques for object detection using web-sourced datasets and offers a large dataset to facilitate further studies.

**Strengths:**

1. The paper is well-written and easy to understand. The inclusion of various figures significantly aids comprehension, providing visual insights into the content.
2. The proposed method is simple and straightforward.

**Weaknesses:**

The paper's approach to data curation for object detection raises some concerns, particularly in terms of its dependency on pretrained models and the potential limitations this might impose.

It is a common understanding that expanding the exploration of the input space through pseudo-labeling a larger dataset and subsequent fine-tuning, akin to practices in domain adaptation or semi-supervised learning, can lead to performance improvements. This process essentially leverages unlabeled data to enhance the model's performance.

However, doubts emerge when considering the scenario where Ground Truth labels are available for a specific subset of images (e.g., 12.8 million images). In such a case, the optimality of the authors' data curation method is questionable, especially since it seems to avoid incorporating challenging images. To elaborate, the proposed method heavily relies on a pretrained model in two significant ways: it filters out 1) images if the pretrained model does not have sufficient confidence in the object proposals, and 2) images where the entropy for the classes used in pre-training is low. Therefore, the proposed approach inherently disregards images containing challenging objects and those with unseen classes, potentially limiting the diversity and richness of the curated dataset.

From the perspective of pseudo Ground Truth, selecting uncertain items by the model would naturally result in lower label quality, leading to no performance gains. However, this argument primarily holds under the continuous use of pseudo GT.

In summary, I perceive the proposed method more as a treatment for issues in semi-supervised learning or domain adaptation, rather than a robust data curation method for object detection. The reliance on pretrained models and the exclusion of challenging or unseen instances could lead to a curated dataset that lacks complexity and diversity, potentially hindering the model's ability to generalize and perform well on a broader range of tasks.

**Questions:**

Please check the Weaknesses

---

### Official Review · Reviewer_MZQa · 2023-11-02

**Soundness:** 2 fair
**Presentation:** 3 good
**Contribution:** 2 fair
**Rating:** 3
**Confidence:** 4

**Summary:**

The paper proposes two methods to select a subset from a larger dataset for object detection training. The two methods are region proposal based and class entropy based. Pseudo-labels are generated by an object detector pretrained on LVIS for the selected subset. A detector is trained on the selected subset and then finetuned and tested on LVIS for final results. The paper claims the proposed data filtering method is more effective than CLIP-based methods.

**Strengths:**

The paper performed experiments at scale and showed results with different scales of selected datasets.
It also demonstrated that using a pretrained objector for data filtering is more effective than using a detection-unaware CLIP model to measure the image and caption consistency.

**Weaknesses:**

The 1% performance increase when scaling from 6.4M to 102.4M somehow showed the proposed method was unpromising for scaling up the dataset, even if it was better than random selection (by ~1%).

The detection accuracy is generated by training a new detector using pseudo labeled generated by a pretrained detector and then finetuning the detector on the same dataset (LVIS) as the pretrained detector trained on. This experimental setting makes everything overfitted to a single dataset. I suspect this strategy can create too much bias to make the experimental results not generalizable to more general scenarios (e.g., finetuning on a different domain).

**Questions:**

It will be helpful to justify the usefulness and significance of the proposed method beyond the ~1% accuracy improvement over using unfiltered DataComp data.